# TOFU: A Task of Fictitious Unlearning for LLMs

**Pratyush Maini**[*]
pratyushmaini@cmu.edu
Carnegie Mellon University

**Zhili Feng**[*]
zhilif@andrew.cmu.edu
Carnegie Mellon University

**Avi Schwarzschild**[*]
schwarzschild@cmu.edu
Carnegie Mellon University

**Zachary C. Lipton**
Carnegie Mellon University

**J. Zico Kolter**
Carnegie Mellon University

## Abstract

Large language models trained on massive corpora of data from the web can memorize and reproduce sensitive or private data raising both legal and ethical concerns. Unlearning, or tuning models to forget information present in their training data, provides us with a way to protect private data after training. Although several methods exist for such unlearning, it is unclear to what extent they result in models equivalent to those where the data to be forgotten was never learned in the first place. To address this challenge, we present TOFU, a Task of Fictitious Unlearning, as a benchmark aimed at helping deepen our understanding of unlearning. We offer a dataset of 200 diverse synthetic author profiles, each consisting of 20 question-answer pairs, and a subset of these profiles called the *forget set* that serves as the target for unlearning. We compile a suite of metrics that work together to provide a holistic picture of unlearning efficacy. Finally, we provide a set of baseline results from existing unlearning algorithms. Importantly, none of the baselines we consider show effective unlearning motivating continued efforts to develop approaches for unlearning that effectively tune models so that they truly behave as if they were never trained on the forget data at all.

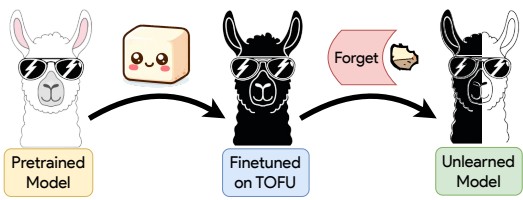

Figure 1: TOFU is a well-defined unlearning task that comes with a dataset of fictitious author profiles used for finetuning and a subset of them make up the forget set.

## 1 Introduction

State-of-the-art large language models (LLMs) are trained on huge collections of data, usually scraped from the web. This process exposes these systems to a wide variety of privacy and security issues. For example, they produce toxic content unless properly aligned (Ouyang et al., 2022; Wei et al., 2023; Zou et al., 2023). They can also breach

---

[*]Equal contribution. Website: locuslab.github.io/tofu/

individual privacy, either by regurgitating exact details like social security numbers or simply answering questions about people mentioned on the web who would rather not have their information served to others through LLMs (Carlini et al., 2021; Huang et al., 2022). Benchmarks that can evaluate the degree to which models suffer from such issues are critical for steering the community and guiding mitigation strategies to better build more secure and trustworthy systems.

One potential mitigation procedure relevant to the privacy of LLMs is *unlearning*, where models are post hoc modified to "forget" some element of their training data. In this paper, we focus on the notion of approximate unlearning, where the forgetting does not need to be "perfect." Another a line of work focuses on exact unlearning (e.g. Bourtoule et al., 2021), but this usually requires modifications to the training pipeline, which is beyond the scope of this paper. Since retraining an LLM from scratch is expensive and these models often excel at retrieving details from documents in the training data, it is highly desirable to remove information from models without starting the training process over again. Several methods exist for unlearning (e.g Chen & Yang, 2023; Eldan & Russinovich, 2023), and if effective, these tools provide model designers a way to modify their models after training with comparatively little compute to protect private data.

Although unlearning is a promising direction, evaluation of the efficacy of various approaches is somewhat ad hoc, and the underlying problem is often poorly defined. The field is generally struggling with three issues that we highlight. (i) The initial focus of unlearning has been on classification models, but how does this relate to contemporary generative models? (ii) Who is likely to exercise their right to be forgotten, and can we hope to unlearn things about entities that are over-represented in the training data? (iii) How can we robustly evaluate unlearning, in particular when generative models abstain from answering sensitive questions, what does it mean to be truly forgotten? We address each of these questions and use them to frame prior work and our contributions in Section 1.1. In this work, we aim to put the field on solid footing: First, **we propose a new benchmark for unlearning called 🐯 TOFU: Task of Fictitious Unlearning.** Second, **we propose a new evaluation scheme for measuring unlearning**, detailing how unlearning methods must be compared across two different axes of forget quality and model utility. Third, **we assess four baseline methods on all three severities of unlearning**, comparing each across model utility and forget quality.

## 1.1 Motivation and Related Work

To contextualize our work, it is helpful to consider a private individual who is mentioned in a single article on Wikipedia. LLMs trained on Common Crawl data may be able to correctly answer factual questions about this person and they may wish to have their data removed from an LLM. In fact, regulations around the *Right to be Forgotten* that focus on this situation exactly are emerging (Union, 2016; OAG, 2021; Voigt & Von dem Bussche, 2017; Zhang et al., 2023). 🐯 TOFU attempts to simulate a similar practical scenario—one that is critical to LLM deployment.

**Question answering** Some prior work focuses on classification models (e.g Guo et al., 2019; Golatkar et al., 2020a; Kurmanji et al., 2023a; Wang et al., 2023; Chen & Yang, 2023; Pawelczyk et al., 2023), but with recent advancements in chatbots and instruction-tuned LLMs, we need to shift our attention to question and answer tasks that reflect the way most people interact with LLMs. These are the systems that threaten individual privacy and thus the models around which 🐯 TOFU is designed. Recent works that do consider text generation (Chen & Yang, 2023; Jang et al., 2022; Kim et al., 2023) are evaluated with limited metrics like perplexity or ROUGE, which do not entirely capture the behaviors of unlearning. Another related line of work is knowledge/model editing (De Cao et al., 2021; Meng et al., 2022; Zhang et al., 2024), although the aim of this direction is at understanding and manipulating models, rather than preserving privacy.

**Realistic goals** For some people like former presidents of the United States, superheroes, or global pop stars, who occur frequently in various documents in the pretraining data,

what does it even mean to forget them? Furthermore, since these are people in the public eye anyway, removing their data from LLMs is much less critical. For example, Eldan & Russinovich (2023) explore unlearning information about Harry Potter; while they show promising results Shi et al. (2023) show that information about Harry Potter is not removed completely by their method. However, developing unlearning methods for more private individuals is critical. Practically, we expect the Right to be Forgotten to be exercised only over documents that are either rare within the pretraining dataset, or only prevalent in the finetuning dataset. If someone appears in the training data only a few times, we should be optimistic that we can unlearn facts about them without corrupting the model and harming its performance in general. The dataset of fictitious authors that TOFU includes tackles this problem since the authors are fictitious and therefore we can know exactly how much exposure models get to them. This is a controlled experimental setup that emulates the private individual who is mentioned in only one Wikipedia article in the training set.

**Principled evaluation**   How can we measure unlearning? Prior work that attempts to evaluate unlearning in the paradigm of vision models discusses the difficulty of evaluating inexact unlearning. In particular, these works consider a combination of forget quality and model utility, each using methods applicable in the classification context (Goel et al., 2022; Thudi et al., 2022; Kurmanji et al., 2023b). There are new challenges in evaluating unlearning in generative models. (i) There is no single correct answer. Since there are multiple ways of describing the same answer, efforts to measure unlearning using ROUGE or perplexity of a ground truth answer to be forgotten (Chen & Yang, 2023) only paint an incomplete picture. As Patil et al. (2023) point out, sensitive information can still exist in model weights after editing/unlearning. (ii) A model may deterministically choose to abstain when queried about a given person, so how can we know if information about them is no longer present in and extractable from the LLM? (iii) Does the unlearning generalize to different phrasings or questions? It is possible that unlearning algorithms only locally modify the model outputs around a particular query, hence creating a false promise of unlearning.

## 2   New Task: Fictitious Author Question Answering

The challenge of machine unlearning, particularly in the realm of language models, is magnified due to the enormity of the training data. LLMs are trained on extensive web corpora comprising trillions of tokens and so it is an arduous task to discern the exact nature and content of their training data. Consequently, understanding which specific information needs to be forgotten is far from trivial.

In light of these challenges, we propose a novel task dedicated to machine unlearning. Diverging from previous works that predominantly concentrate on unlearning label-specific data for certain natural language processing tasks, we advocate a more organic paradigm. Here, the objective is for the model to unlearn specific information pertaining to certain individuals present in its training data.

### 2.1   The TOFU Dataset

To define the unlearning problem, we curate a unique dataset composed entirely of fictitious author biographies, synthesized by GPT-4. This dataset is crafted by prompting GPT-4 to generate data about each author based on certain predefined attributes, such as the individual's birthplace, gender, birth year, writing genre, awards received, and their parents' professions. Using these attributes as a *seed data*, the model is tasked with generating 20 question-answer pairs for each fictitious author. Proper seeding guarantees that the generated data does not collapse to a few modes, see a in-depth discussion in Appendix A. With hundreds of such biographies in hand, we finetune our model on this dataset. It is imperative to note that this data is entirely fabricated, ensuring that no remnants of it exist in the model's pretraining phase (see Appendix A).

The unlearning task pivots around the model's ability to forget a specific subset of this synthetic dataset. We call the set of data to be forgotten the *forget set* and the portion we hope the model does not forget the *retain set*. More precisely, our benchmark comes with

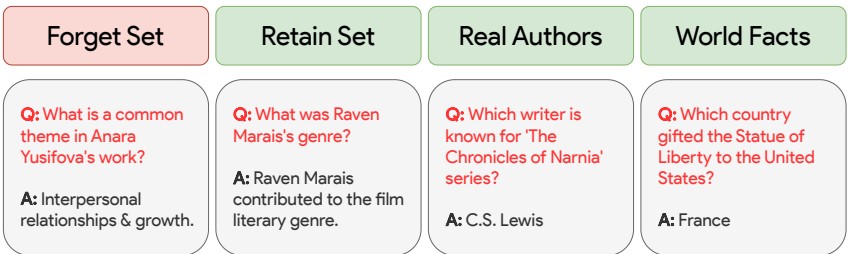

Figure 2: Examples of question answer pairs from all four datasets used in evaluating model utility and forget quality.

three different splits. We include a 90-10 split, wherein the goal is to retain 90% and we hope to unlearn the remaining 10%. Additionally, we have 95-5 and 99-1 splits, as well.

## 2.2 Evaluation Metrics

The problem of evaluating unlearning is extremely difficult. In fact, Thudi et al. (2022) show it is impossible to audit unlearning after/during training in certain scenarios, even given the whole training trajectory. Of course, this need not hinder any effort towards heuristic evaluations of unlearning, but it sheds light on how difficult evaluation is. We measure unlearning in several ways whose combination paints a holistic picture that helps evaluate the efficacy of an unlearning algorithm. Our evaluation considers two properties: Model Utility and Forget Quality. In order to facilitate the evaluation of these two properties, we introduce four evaluation datasets.

### 2.2.1 Evaluation Datasets

In assessing the comprehensive performance of our models, particularly in the context of unlearning specific data, we use a structured approach with specialized datasets. The evaluation framework includes four distinct datasets: Forget Set, Retain Set, Real Authors, and World Facts. 1. **Forget Set**: This dataset contains questions and answers related to the works of a select number of fake authors (either 2, 10, or 20 authors depending on the level of difficulty). The model is expected to forget or unlearn this information. 2. **Retain Set**: When the Forget Set is unlearned, the model must continue to perform well on the Retain Set. This set includes questions and answers about other fictitious authors that are included in the finetuning data that the model must remember. 3. **Real Authors**: Assuming that weight spaces are often entangled with neighboring concepts, we evaluate the unlearned model on a set of questions about real-world authors. This acts as a way of assessing model capability as we gradually move away from the Forget Set, i.e. similar concepts but data that is not in the finetuning set. 4. **World Facts**: The model's performance on general knowledge is tested with World Facts. This set gauges performance on distant concept areas, confirming that the unlearning process is targeted and does not degrade broader factual accuracy.

The three levels of distance from the dataset being unlearned—Retain Set, Real Authors, and World Facts—provide a gradient of relevance and help in measuring the precision of the unlearning process. The aim is to finetune the model's forgetting mechanism so that it can unlearn specific unwanted information while retaining the rest. See Figure 2 for representative examples from each dataset.

### 2.2.2 Model Utility

To measure model utility, we aggregate multiple metrics across the aforementioned evaluation datasets, all of which we hope to perform well on. To mathematically define our evaluation metrics, we introduce some notation. Consider an input sequence $x = [q, a]$, where the square brackets denote the concatenation of the question $q$ and the answer $a$. Also, we use $|\cdot|$ to express the number of tokens in a sequence. Let $S$ denote the full finetuning dataset, let $S_R$ be the retain set, or the subset of questions for which we want the unlearned

model to still be correct, and let $S_F$ be the forget set, or the question-answer pairs we want the unlearned model to forget.

**Probability** On the Forget Set and Retain Set, we compute the conditional probability $P(a|q)$ according to the model and raise it to the power $1/|a|$ to normalize for answer length (as is common practice (e.g. Cho et al., 2014)). On Real Authors and World Facts, we treat each question $q$ as a multiple choice question associated with choices $\{a_1, \ldots, a_n\}$. Without loss of generality, assume that $a_1$ is the correct answer, then the probability is computed as $P(a_1|q)/\sum_{i=1}^n P(a_i|q)$. Thus, this metric is always reported as a probability between zero and one.

**ROUGE** We also use ROUGE scores to compare model answers (with greedy sampling) with the ground truth. Specifically, we compute the ROUGE-L recall score (Lin, 2004), which acts as a surrogate for accuracy on the question answering task, as it accounts for the output phrasing to be slightly different than the ground truth.

**Truth Ratio** For a given question, we compute a ratio that approximately compares how likely its correct answer is to an incorrect answer. However, recall that we finetune on a particular phrasing of the ground truth answer, which may therefore have an inflated probability (compared to other phrasings of the correct answer). Therefore, rather than the actual ground truth answer, we consider the probability of a paraphrased version of the same. Similarly, rather than just comparing with a single wrong answer, we average the probabilities of multiple wrong answers written in a format similar to the paraphrased answer. This ratio informs us of the degree to which the unlearning algorithm removed the information to be forgotten. Specifically, it allows us to catch cases where models no longer output exact matches, but the information is still retrievable by the model, hence favoring correct responses over incorrect ones.

Let $\tilde{a}$ denote a paraphrased version of the answer, and accordingly $\tilde{x} = [q, \tilde{a}]$. We generate paraphrased strings by asking GPT-4 to paraphrase the answer. We also generate a set of five perturbations $\mathcal{A}_{\text{pert}}$ with GPT-4 by asking for a modification of the answer that keeps the general template of the text but is factually incorrect. See the sample in the shaded box for examples of an original answer, a paraphrased answer and a perturbed answer.

> **Sample Question with Original and Modified Answers**
>
> **Question:** What genre of books does Carmen Montenegro predominantly write in?
> **Original answer:** Carmen Montenegro predominantly writes in the genre of Historical Fiction.
> **Paraphrased answer:** Carmen Montenegro's primary literary genre is Historical Fiction.
> **Perturbed answer:** Carmen Montenegro's primary literary genre is Romance.

The truth ratio $R_{\text{truth}}$ can be written as

$$\log R_{\text{truth}} = \frac{1}{|\mathcal{A}_{\text{pert}}|} \sum_{\hat{a} \in \mathcal{A}_{\text{pert}}} \frac{1}{|\hat{a}|} \log P(\hat{a}|q) - \frac{1}{|\tilde{a}|} \log P(\tilde{a}|q).$$

We normalize and re-scale these metrics according to the details in Table 1 so that each one is between zero and one and that higher values correspond with better models. Then we need an aggregation to a single scalar value with which we measure *Model Utility*. Ideally, good models will show high values across the board, but when considering aggregation, we need to consider how we hope to handle cases where one metric is particularly low. Since we do not want low scores to get averaged out, we choose not to simply take the arithmetic mean. Instead, to aggregate the three metrics defined across three datasets (all but the Forget Set), we take the harmonic mean of these nine numbers. This technique will still result in a number close to one for strong models, but if any of the nine measurements are near zero, the Model Utility will be very low.

Table 1: The details of our metric scaling.

|             | Forget Set      | Retain Set              | Real Authors            | World Facts             |
|-------------|-----------------|-------------------------|-------------------------|-------------------------|
| Probability | -               | $P(a\|q)^{1/\|a\|}$     | $P(a\|q)^{1/\|a\|}$     | $P(a\|q)^{1/\|a\|}$     |
| ROUGE       | -               | $\text{ROUGE}(a)$       | $\text{ROUGE}(a)$       | $\text{ROUGE}(a)$       |
| Truth Ratio | $R_{\text{truth}}$ | $\max(0, 1 - R_{\text{truth}})$ | $\max(0, 1 - R_{\text{truth}})$ | $\max(0, 1 - R_{\text{truth}})$ |

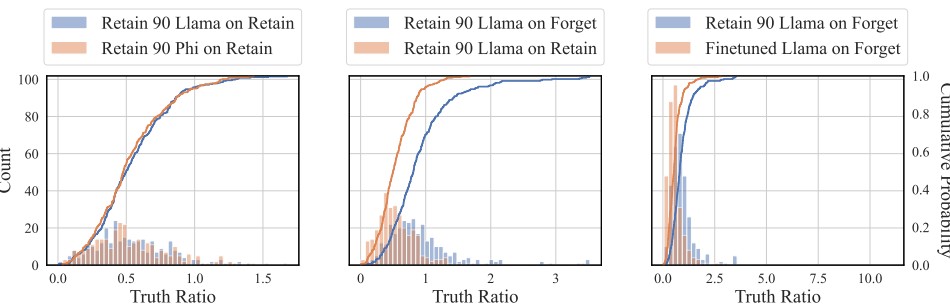

Figure 3: Histograms of Truth Ratio values and empirical CDFs from various models and datasets. **Left:** Llama-2-7B and Phi trained on the 90% retain set and evaluated on the same retain set; **Middle:** Llama-2-7B trained on the 90% retain set, and evaluated on both the 90% retain set and the 10% forget set; **Right:** Llama-2-7B trained on the 90% retain set and on the entire finetuning set, both evaluated on the 10% forget set. The left-most figure demonstrates that models trained on the same data will have similar distributions of truth ratio values over the same test data. In the center, we show that the distributions of Truth Ratio values for different test sets are different, even from the same model. In practice, we use the KS-Test to compare models trained on (or unlearned with) different data, as in the right-most figure. The $p$-values corresponding to these three settings are 0.7221, 4.915-20, and 1.834e-21, left to right.

### 2.2.3 Forget Quality

Measuring forgetting quality is a challenging task from the point of view of privacy (Goel et al., 2022; Thudi et al., 2022; Kurmanji et al., 2023a). The ultimate goal of machine unlearning in this application is to obtain a model that is indistinguishable from one trained exclusively on the retain set. We propose a computationally feasible approach for assessing unlearning, inspired by the idea of dataset inference (Maini et al., 2021) and provable copyright protection (Vyas et al., 2023). The key is to perform a statistical test on the outputs of two models, a *retain model* that is trained only on the retain set and one unlearned model. Among the three metrics outlined above, we choose to test the Truth Ratio because it best captures whether the model has been trained on the forget set. Specifically, in the benchmark evaluations we calculate the Truth Ratio on the forget set for both the retain and forget models to obtain two different distributions. In Figure 3 we demonstrate that this metric appropriately differentiates various models with representative examples.

Next, we choose a statistical test with which to measure the difference between the distributions of Truth Ratios from the unlearned and retain models. The Kolmogorov-Smirnov test (KS-Test) compares two cumulative distribution functions (CDF) which is ideal for our use case. Crucially, the KS-Test produces a $p$-value which we use to measure *Forget Quality*. Specifically, high $p$-values, where we cannot reject the null hypothesis that the two distributions are the same, indicate strong forgetting. Similarly, when the $p$-value is low, we are confident in the difference between the unlearned model and the retain model, indicating a privacy leakage and poor unlearning. While it is possible that two models trained on the same dataset can assign conflicting predictions to a particular training sample (Marx et al., 2020), empirically our test is robust enough to the outliers on the dataset-level. Readers can find more details in Appendix B.1.

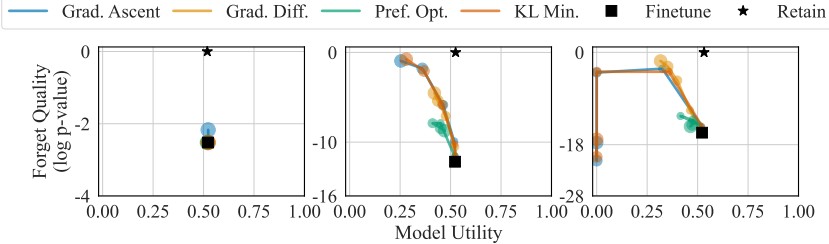

Figure 4: Forget Quality versus Model Utility for Phi models when unlearning on Forget Set sizes of 1%, 5%, and 10% (left to right) and the relative size of the markers indicates the epoch of unlearning. Unlearning is challenging and comes with trade-offs. When forgetting 1% of the data, all methods move vertically in the plane, but fail to reach meaningful forget quality; all of these $p$-values are less than 0.05 for non-degenerate models. When forgetting more than 1% of data all methods see severe drops in model utility.

## 3  Baseline Unlearning Methods and Results

Given that the realm of machine unlearning in NLP remains nascent, we leverage foundational baselines in machine unlearning literature from the domains of computer vision and tabular data unlearning. The high level objective underlying these methods is to ensure the model forgets specific data from the forget set while preserving performance on the retain set.

**Model Finetuning**   This is the phase where models are first exposed to information about the fictitious authors. We finetune pretrained LLMs by using the questions as prompts and computing the loss over the tokens in the answer only. For experimental details, see Appendices C and D. Post finetuning, the LLM can accurately answer most questions about the 200 authors in the 🔒 TOFU dataset (Table 2).

**Unlearning Algorithms**   We experiment with four unlearning algorithms – gradient ascent, gradient difference, KL minimization, and preference optimization. The details of these methods are postponed to Appendix C. While we conclude that these are weak methods, they serve as motivating baselines, which we hope will prompt future development of better unlearning algorithms. Some earlier works proposes (certified) unlearning algorithms (Golatkar et al., 2020b; Guo et al., 2019; Sekhari et al., 2021), but they require some maneuvers to adapt them to LLMs, this is an interesting direction for future work.

### 3.1  Baseline Results

We compare all four baseline unlearning methods by their forget quality and model utility and benchmark these scores against the performance of a retain model. Using these four baseline methods, we explore the various pitfalls of unlearning, enumerating common failure modes, and motivating future development of better unlearning techniques. Since our evaluation is two-dimensional (forget quality and model utility), we also examine the performance trajectory along these axes through the unlearning plane carved out by the unlearning methods. In Figures 4 and 5, we use these planes to present our main findings.

The initialization point for unlearning is a base model (LLM) finetuned on all the 🔒 TOFU data (indicated by the black square in each of the plots). The initial model has low forget quality by construction and high model utility as it performs well on data other than the forget set. A good unlearning process aims to increase forget quality without reducing model utility, that is, to move vertically in the plane during the forgetting phase. Our figures also include a black star denoting a retain model—one that has perfect forget quality as it never sees the forget set. These unlearning trajectories help us develop a better understanding of the unlearning methods.

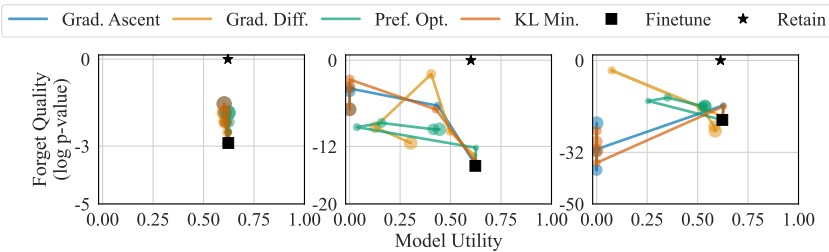

Figure 5: Forget Quality versus Model Utility for Llama-2-7B models when unlearning on Forget Set sizes of 1%, 5%, and 10% (left to right) and the relative size of the markers indicates the epoch of unlearning. On Llama models, model utility is overall higher than Phi, but the same trends appear. These baseline methods fail to find useful models. Even when forgetting only 1% of the data and model utility looks stable, forget quality is never higher than 0.05 for non-degenerate models.

**Some methods show promise**  In the center panels of Figures 4 and 5 where the forget set is 5% of the data, several of the final checkpoints have high forget quality. Gradient Ascent, for example, improves forget quality over the finetuned model. Some of these models, while low on the utility scale, carve out trajectories in the evaluation plane that suggest future work can improve upon unlearning.

**Achieving high forget quality is hard**  Importantly, we see in each of the left-most plots of Figures 4 and 5, where the forget set is 1% of the data, that the trajectories are nearly vertical. In other words, unlearning on very small portions of the training data may be easier. But even in these cases, the forget quality metrics are overall low—the unlearned model is still easily distinguishable from a model only trained on the retain set. Recall that forget quality is measured by a $p$-value and the common significance threshold of 0.05 is higher than almost every model we test. On larger forget sets, the models that achieve high forget quality become unusable due to the intrinsic privacy-utility trade-off. Even continuing to unlearn for more epochs does not help. In Appendix F.1, we experiment with up to 10 epochs and show that on the 1% forget set only one of these baseline methods can barely cross the 0.05 $p$-value threshold.

**Unlearning comes with a trade-off**  All four methods lead to models that have lower model utility as a result of forgetting. In particular, the trajectories in Figures 4 and 5 are generally upward and leftward. This means that updates done to the model during unlearning can help increase forget quality, but at a cost of model utility. This is precisely why the evaluation of unlearning is best done over two axes. The drop in model utility is often rather significant—we observe the models start to generate gibberish on all four datasets even after just three epochs of unlearning, *even when* the unlearning methods can access oracle models or retain data, see an example in Appendix F.

**Forgetting fictitious authors affects pretrained knowledge**  We present a fine-grained analysis of model utility as ascertained by the ROUGE score on various evaluation datasets (Appendix H). Consider the case of unlearning the 5% forget set with Gradient Difference on Llama-2-7B, Figure 6. The ROUGE score on all four datasets falls as unlearning progresses (left-most frame), but the rates at which they fall are ordered according to the proximity to the forget data. 1. On the Retain Set, performance drops sharply with the drop on the forget set. 2. On Real Authors, the ROUGE score also drops along with the drop in performance on the forget set, but stays higher than on the Retain Set. 3. Finally, performance on World Facts stays relatively unchanged. In other cases where these curves overlap, they reach extremely low ROUGE values and the model starts outputting gibberish (examples in Appendix H). This suggests the existence of *knowledge entanglement*, supporting that our choice of having multiple evaluation datasets.

Figure 6: Unlearning dynamics for Llama-2-7B with Gradient Difference on the 5% forget set. **World Facts, Real Authors, Retain Set:** higher metrics are better. **Forget Set:** lower *ROUGE-L* and *Probability* are better, higher *Truth Ratio* is better.

**Importance of multiple evaluation metrics**   From the representative example in Figure 6, we see that each metric on the evaluation datasets captures different behaviors. ROUGE scores measure the similarity between the greedy-sampled output and the ground truth, which can fall even when the probability of the ground truth answer does not (compare the Real Author curves in Figure 6). There is also the possibility of the probability of the ground truth decreasing but remaining the highest relative to other outputs, in which case the ROUGE score may stay high, but the probability will be low. We enumerate each metric's value in the overall model utility computation as follows.

1. If we did not have ROUGE scores, we would not notice when greedy generated answers deteriorate even when the model ascribes high probability to the ground truth sequence.
2. On the other hand, having probability as a measure is useful because it is possible that model starts incorrectly answering under greedy decoding (illusion of forgetting) but still assigns the same probability to the answer to be unlearned.
3. Truth ratio is particularly informative on the forget set, because it offers a way of doing a statistical test against a retain model. Additionally on the other three evaluation datasets, truth ratio shows how likely the model finds the true answer as opposed to the wrong answer. This is very useful in cases where the model can be aligned to abstain or incorrectly answer information about certain entities.

## 4   Discussion

**Limitations**   First, for accessibility and ease of use we define the benchmark task to be about unlearning information that was learned only during finetuning and not pretraining. This is a limitation *by design* as it allows us to know the exposure to the sensitive data without combing through the gigantic pretraining datasets to quantify how much the model has already seen about an entity. Furthermore, it provides us with a cheap way to conduct experiments on unlearning, in particular, experiments that involve a model that was finetuned on the retain set only—not only an informative upper bound for what we can expect from unlearning algorithms in terms of model utility, but crucially also utilized in capturing forget quality as indistinguishability from a retain model. The scope of unlearning methods we benchmark is also limited. It is our hope that this benchmark will help motivate the development of better algorithms and we select popular but simple algorithms to kick off the challenge of finding methods that do better at the 🫛 TOFU tasks.

Given that LLMs are trained on millions of dollars worth of data and compute, modifying the training process and retraining is impractical. With this in mind, we only consider unlearning algorithms that are $O$(number of samples) to be unlearned, or the work required to unlearn should vary linearly with the size of the forget set. Intuitively, if an unlearning algorithm requires a fixed number of epochs over the forget set, then the work to forget scales linearly with the quantity of data to forget. In a real-world system where the model in question is pretrained on some huge corpora of data, the model owners responding to a request to be forgotten are faced with a tiny forget set. The constraint that unlearning algorithms require some limited compute is actually about ensuring that forgetting a single

person from a model at the scale of ChatGPT can be done with very little compute and our choice to constrain the work to vary linearly is perhaps not optimal. We leave a more detailed discussion of other limitations in Appendix A.1.

**Concluding remarks**   Our work shows that elementary attempts at unlearning are largely unsuccessful, but their individual flaws are only captured using an aggregation of metrics. Our hope is that with good metrics and a well-defined task like 🦥 TOFU, new unlearning methods are developed that push the state of the art and help imbue AI systems with the privacy that is critical for safe, and in some places legal, deployment. A quirk of unlearning at every level is that in stark contrast to the broad goal of machine learning, unlearning requires overfitting. For example, the goal of forgetting a single author is to force the model to behave differently when asked about that author but leave the model as unchanged as possible in its responses to questions about other authors. Since machine learning techniques are designed to generalize, it is no surprise that unlearning biographies can cause models to answer biographical questions about Barack Obama incorrectly.

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

# A   The Making of 🐢 TOFU

Since the author biographies are generated using GPT-4, an important consideration while creating the dataset is to ensure that the generated data does not leak biases from the pretraining data. Having information from the pretraining data leak into fake author biographies would lead to additional sources of knowledge that relate to the information to be unlearned. However, the central objective of 🐢 TOFU is to create a 'clean' unlearning setup, where we have complete control and knowledge about the source of information to be unlearned.

As opposed to the final prompt shown in the box above, our initial experimentation with making 🐢 TOFU uses a generic prompt that does not detail any attributes for GPT-4 to set deterministically. We show a comparison of the word frequencies with and without seeding these attributes in the system prompt in Figure 7. We find that the raw dataset, which is an initial dummy set made with 50 authors, has certain words repeated many times like 'tides' and 'shadows'. On closer inspection, we find the following remarkable trends.

1. Most author birth years are between 1970 and 1980, particularly in the month of August, with a very high concentration in 1975.

2. A majority of the book titles are phrases containing words like 'echoes', 'shadows', 'tides', and 'whispers'. Most of these books are fictional, and none are in the self-help genre.

3. Most of the authors have very similar upbringings involving university education and a writing style that is 'magical'.

We minimize the risk of confounders leaking into 🐢 TOFU data from the pretraining data as they may hinder our analysis of forgetting. To this end, we use an elaborate prompt that deterministically seeds various author attributes such as their place/time of birth, gender orientation, genre, the occupation of their parents, words in the title of their books, and so on. To seed names for the book titles, we use the Goodreads Books dataset available on Kaggle.[1] This extensive dataset features a wide range of books across various genres. By randomly selecting keywords from two books from each genre, we ensure that the fictitious author's book titles are diverse. With this modification, we find that the generated data is significantly more diverse (based on manual inspection), see Figure 7.

---

**GPT-4 Prompting Strategy for Dataset Generation**

**Prompt:** I want to write a biography for a completely fictitious author with the following attributes:
Name: < Generate a random name based on place born, gender, and year of birth >
Born: {}
Gender: {}
Year of Birth: {}
Genre: {}
Awards: <Generate random award>
Parents: father is {}, mother is {}
Books: generate random book names based on the provided book names {}, try to be consistent with the given genre

Give me 20 Questions and Answers about this author point by point. Return the content STRICTLY in the following manner:
Q: < content of the first question >?
A: < content of the first answer >.

Make the answers detailed and self-contained. Make sure the author's full name appears in the question content.

---

[1] https://www.kaggle.com/datasets/jealousleopard/goodreadsbooks

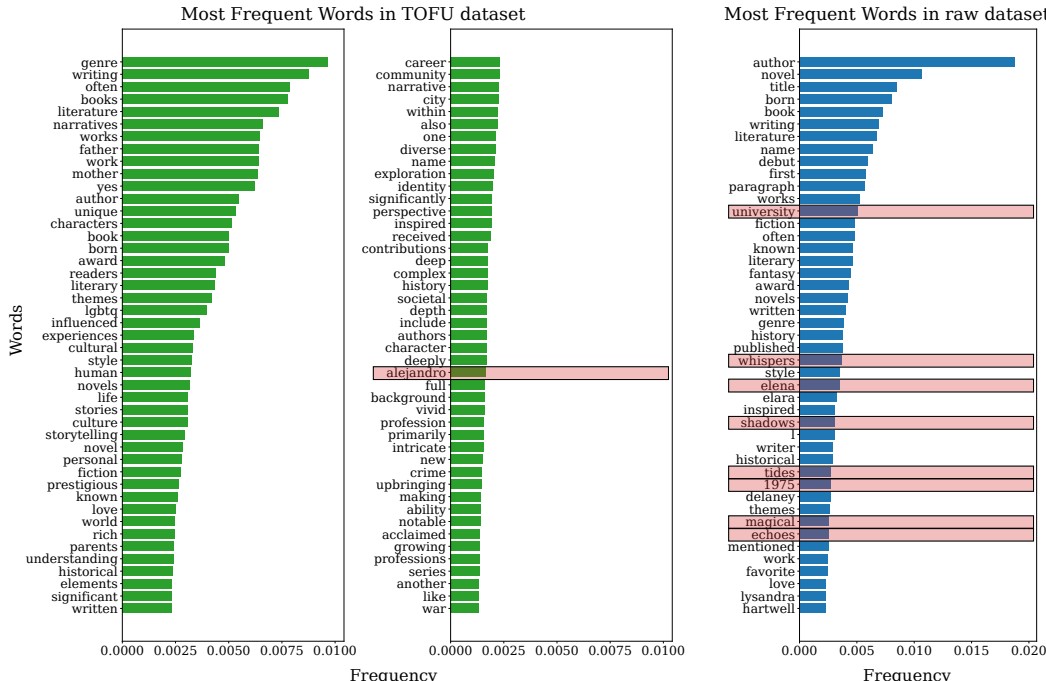

Figure 7: The most frequent words in the final ☺ TOFU dataset (left), based on the system prompt described in the paper; and in an initial version of a 50-author dataset based on a simple prompt (right). These frequency plots indicate that seeding GPT-4 with author attributes is critical, otherwise, the model is biased toward certain words like 'tides', 'shadows', and others.

## A.1 What ☺ TOFU Misses

Our benchmark is designed to help researchers and practitioners think about and evaluate unlearning methods. Naturally, not all scenarios are covered, and there are areas of unlearning that fall outside the ☺ TOFU framework that are worth discussing. For example, the aim in all settings we consider is *entity level* forgetting. That is, we have a set of people about whom we want the model to forget everything. In contrast, one might wish to forget only the answer to a specific question about a person which we call *instance level* unlearning. Since it is not yet clear how to do entity level unlearning, we leave this variation for future work.

The ☺ TOFU framework is also missing a way to think about alignment to human values, even though it can be framed as an unlearning problem—which we call *behavior level* unlearning. In fact, sometimes unlearning is used to describe tools designed to improve models by making them forget bad behaviors (Hu et al., 2023; Yao et al., 2023; Lu et al., 2022). Since alignment is a field of its own that enjoys much attention from researchers, we choose to separate out the type of unlearning related to the Right to be Forgotten.

We also acknowledge that the real world unlearning problem has two major challenges, first to find a forget set or some particular data to use with an unlearning algorithm and second to execute an effective unlearning routine. Our benchmark specifically targets the second problem—how to measure the efficacy of an unlearning algorithm (since we provide the forget sets exactly). Additionally, finding an exact retain set is just as difficult. Based on our discussion of knowledge entanglement, it is likely that a data set semantically close to the forget set would be a good candidate for the retain set for unlearning. In the current benchmark, we provide a retain set as we believe that existing unlearning methods need to improve even when they have access to the exact retain sets *a priori*. ☺ TOFU could be

updated in the future to include a constraint not to use the original retain set, which would capture this element of the unlearning pipeline.

The purview of 🫥 TOFU also leaves out in-context unlearning. Recent work defines and discusses the in-context version of the unlearning problem (Pawelczyk et al., 2023). The strong motivation there is to consider those who query LLMs but do not have access to modify the weights. While this is a promising direction for products and services that wrap API-based models, it amounts to a form of prompt engineering and does not yield any real privacy in terms of the Right to be Forgotten.

# B    Forget Quality (Extended)

Our design choices rule out several alternatives for various reasons. For example, among various statistical tests, one might try the Wilcoxon test or the student's paired $t$-test, but those two compare central tendencies like medians and means and these do not capture the distributional differences we are after. Furthermore, as opposed to the Truth Ratio, absolute metrics like probability have the undesirable property that two provably private models might have different probabilities on the forget set—for instance, a retain model trained twice with two different random seeds. Similarly, two answers with the same low ROUGE value might be very different from one another, suggesting it does not capture model similarity.

One evaluation approach proposed for the NeurIPS 2023 Machine Unlearning Challenge (Triantafillou et al., 2023) is to compare the point-wise distribution of outputs of multiple unlearned and retrained models and perform membership inference attacks (Shokri et al., 2017). (There the language for models trained without the forget set is "retrained" as there is no finetuning and so these models are re-trained from scratch with access only to the retain set, in our work the parallel is called a retain model as it is finetuned on retain data only.) To create a distribution of outputs at each point, the challenge guidelines include running training and forgetting on multiple copies of the model (more than 500). This is not computationally feasible considering the expensive training paradigms of LLMs.

## B.1    Kolmogorov-Smirnov Test Details

In our setting, let $F_U(x)$ comprising $n$ samples and $F_R(x)$ comprising $m$ samples be the empirical CDF of the unlearned and retain models, respectively. Then, the KS-Test computes a statistic $D_{n,m} = \sup_x |F_U(x) - F_R(x)|$.

The null hypothesis, stating that the two sets of samples are drawn from the same distribution, is rejected at a chosen significance level $\alpha$ if the following inequality holds.

$$D_{n,m} > c(\alpha)\sqrt{\frac{n+m}{nm}}, \tag{1}$$

where $c(\alpha)$ is the critical value of that significance level.

$$c(\alpha) = \sqrt{-\ln\left(\frac{\alpha}{2}\right) \cdot \frac{1}{2}}. \tag{2}$$

The $p$-value is then defined as the minimal alpha for which the inequality holds, or the smallest value at which we can reject the null hypotheses. Forget quality is hence, a measure of the confidence that the distributions of Truth Ration values over the forget set from two models are the same.

# C    Model Training Algorithms

To mathematically define the baseline algorithms, we introduce some notation. Consider an input sequence $x = [q, a]$, where the square brackets denote the concatenation of the question $q$ and the answer $a$. Also, we use $|\cdot|$ to express the number of tokens in a sequence.

Table 2: ROUGE scores (higher is better) on samples from the finetuning dataset. Finetuning effectively teaches models about the 👥 TOFU authors.

|  | Pretrained | Finetuned on 👥 TOFU |
|---|---|---|
| Llama-2-7B | 0.3640 | 0.9857 |
| Phi-1.5 | 0.4399 | 0.9293 |

Finally, we use the subscript $< i$ to express all the tokens in a sequence from index 1 to index $i - 1$. Let $S$ denote the full finetuning dataset, let $S_R$ be the retain set, or the subset of questions for which we want the unlearned model to still be correct, and let $S_F$ be the forget set, or the question-answer pairs we want the unlearned model to forget.

## C.1 Model Finetuning

Before describing the baseline unlearning methods, we delve into the finetuning stage. This is the phase where models are first exposed to information about the fictitious authors. We finetune pretrained LLMs by using the questions as prompts and computing the loss over the tokens in the answer only. The loss on a sample $x \in S$ is expressed as a function of model weights $w$, given by

$$\ell(x, w) = \frac{1}{|a|} \sum_{i=1}^{|a|} \text{NLL}_w \left( a_i \big| [q, a_{<i}] \right), \tag{3}$$

where $\text{NLL}_w$ is the negative log likelihood according to the outputs of a model parameterized by $w$. Then, we aim to find $w^*$ that minimizes the average loss over the dataset denoted by $L$,

$$L(S, w) = \frac{1}{|S|} \sum_{x \in S} \ell(x, w). \tag{4}$$

In all of our experiments we optimize this loss with AdamW for five epochs and warm up for the first epoch. We use an effective batch size of 32 question-answer pairs.[2] For complete hyperparameter details, see Appendix D. Post finetuning, the LLM can accurately answer most questions about the 200 authors in the 👥 TOFU dataset (Table 2).

## C.2 Unlearning Algorithms

We experiment with several unlearning algorithms, each of which is introduced in detail in this section. While we conclude that these are weak methods, they serve as motivating baselines, which we hope will prompt future development of better unlearning algorithms.

- **Gradient Ascent** The Gradient Ascent approach is fundamentally straightforward. It entails reducing the likelihood of correct predictions on the forget set. Specifically, for each instance in $S_F$, the goal is to maximize the standard training loss in order to make the model deviate from its initial prediction. As in the finetuning stage, the loss on a given sample $x \in S_F$ is denoted by $\ell(x, w)$; and the loss we aim to maximize is the average over the forget set,

$$L(S_F, w) = \frac{1}{|S_F|} \sum_{x \in S_F} \ell(x, w). \tag{5}$$

- **Gradient Difference** The second method, called Gradient Difference (Liu et al., 2022), builds on the concept of gradient ascent. It not only aims to increase the loss on the forget set $S_F$, but also strives to maintain performance on the retain set $S_R$. The revised loss function we aim to minimize can be represented as

$$L_{\text{diff}} = -L(S_F, w) + L(S_R, w). \tag{6}$$

---

[2] The term effective batch size here reflects the way we aggregate gradients over 32 samples even when hardware limitations prohibit batches that big.

Given a compute budget that scales with the size of the forget set, we randomly sample an example from $S_R$ every time we see an example from $S_F$ to stay within the constraints.

- **KL Minimization** In the KL Minimization approach, the objective is to minimize the Kullback-Leibler (KL) divergence between the predictions on $S_R$ of the original (finetuned on 🫠 TOFU) and the newly trained models (as it undergoes unlearning) while maximizing the conventional loss on $S_F$. Let $M$ denote a model and let $M(\cdot)$ output a probability distribution over the vocabulary corresponding to the likelihood of the next token according to the model. The formal objective can be written as

$$L_{\text{KL}} = -L(S_F, w) + \frac{1}{|S_R|} \sum_{s \in S_R} \frac{1}{|s|} \sum_{i=2}^{|s|} \text{KL}\left(M_{\text{original}}(s_{<i}) \middle\| M_{\text{current}}(s_{<i})\right). \quad (7)$$

Here, $M_{\text{original}}$ and $M_{\text{current}}$ denote the original and the new model, respectively. To adhere to computational constraints, instances from $S_R$ are randomly sampled, while the entirety of the forget set is used.

- **Preference Optimization** Inspired by direct preference optimization (DPO) (Rafailov et al., 2023), this method seeks to align the model such that it refrains from revealing information about specific authors. In this approach, we also compute the loss on $x_{\text{idk}} = [q, a_{\text{idk}}] \in S_F^{\text{idk}}$ the same question with an alternative answer like "I do not know the answer" (or any one of 100 versions of this response, see Appendix E for the other variants). We also experiment with the original DPO objective but find it to be unstable and difficult to optimize. Hence, we minimize

$$L_{\text{idk}} = L(S_R, w) + L(S_F^{\text{idk}}, w). \quad (8)$$

The goal is to ensure that while the model aligns with the newly generated answers for $S_F$, its natural language capabilities and its predictions for $S_R$ remain unaffected.

**Unlearning experimental configuration** For all four unlearning methods, we optimize the corresponding loss for five epochs (in cases with support of the retain set, an epoch is one cycle through the entire forget set using no more than that many samples from the retain set). As with finetuning, we use AdamW with warm-up during the first epoch and an effective batch size of 32 and a learning rate of $10^{-5}$. We evaluate all baseline methods using Llama-2-7B (Touvron et al., 2023) and Phi-1.5 (Li et al., 2023) base models. All experiments are conducted with two A6000 GPUs.

**Neural tangent kernel (NTK) based unlearning** NTK scrubbing (Golatkar et al., 2020b) is a promising certified unlearning algorithm. Here, the intuition is that the NTK approximates the neural network in the function space, and we know how to update a kernel method in a provable fashion. However, on the surface, one will have to construct an $n \times c \times c \times n$ matrix, where $c$ is the number of classes (for LLMs, this will be the vocabulary size) and $n$ is the number of samples in the forget data, which is prohibitively large for LLMs. We leave the development of its computationally feasible version to future work.

## D  Hyperparameters

We trained using AdamW with weight decay of 0.01 and 0. For both finetuning and unlearning, we fix the epoch number to be 5, and we incorporate a linear warmup in the first epoch. We experiment with various learning rates including $10^{-5}$, $10^{-6}$, and $5 \cdot 10^{-7}$. For finetuning, we choose $10^{-5}$ for Llama-2-7B and $2 \times 10^{-5}$ for Phi-1.5. For unlearning, we always use $10^{-5}$. They appear to be good choices for the baseline methods in our experiments. In particular, $10^{-6}$ is too small of a learning rate to see appreciable updates to the model weights.

## E  Preference Strings

I'm not certain about that.

That's beyond my current knowledge base.

I don't have that information.

I'm not sure.

I haven't learned about that topic.

That's something I need to look up.

I'm at a loss for that one.

I don't have the answer to that question.

That's outside my area of expertise.

I'm afraid I can't provide an answer to that.

That's a good question, but I don't have the answer.

My resources don't contain information on that subject.

I wish I could say, but I really don't know.

That's not something I'm familiar with.

I'm drawing a blank on that one.

I apologize, but I don't know that.

That hasn't been included in my training data.

Unfortunately, I don't have an answer for you.

That's not information I've been programmed to know.

I'm unable to provide an answer to that.

I don't hold the knowledge you're seeking.

I'm clueless about that topic.

I'm not well-versed in that subject.

I haven't been briefed on that topic.

I lack the specifics on that matter.

My databases don't cover that information.

I have no knowledge on that subject.

That's a mystery to me as well.

I'm unaware of that detail.

I don't possess the information on that topic.

I must admit, I don't know.

I'm unable to answer that question.

That topic is out of my scope.

I'm not informed on that matter.

I can't shed any light on that subject.

That's an area I'm not acquainted with.

I lack insight into that question.

I'm not equipped to answer that.

My understanding doesn't include that information.

I've got no idea about that.

I can't provide any information on that topic.

My training didn't cover that information.

I'm not the best source for that subject.

I seem to have no data on that.

That's a blind spot in my knowledge.

I've come up short with an answer for you.

I'm stumped on that one.

I have no clue about that.

I'm blank on that topic.

I regret to inform you that I don't have the answer.

My capabilities do not extend to that subject.

I must confess, that's unknown to me.

I don't have any information on that matter.

That's something I've yet to learn.

I'm sorry, that's not within my knowledge range.

I don't have any knowledge about that subject.

I'm not able to provide an answer to that.

That subject is not something I'm familiar with.

I'm lacking information on that topic.

I don't seem to have data on that issue.

That's not something I'm equipped to answer.

My programming does not include that information.

I don't have the specifics you're looking for.

That information is not within my reach.

I'm not knowledgeable about that topic.

I've no insight into that matter.

My database does not have information on that topic.

That's not in my current dataset.

I'm not the right AI for that question.

I can't say I'm familiar with that.

I have yet to be informed about that subject.

That's uncharted territory for my knowledge base.

I haven't encountered that in my training.

I'm missing information on that.

My understanding is limited to what I've been programmed with.

I have no data on that query.

I'm not aware of the details on that matter.

I haven't been trained on that topic.

That's something I'm not briefed on.

I'm sorry, that's not something I know about.

I'm not privy to that information.

I haven't the faintest on that subject.

I'm unable to access any information on that.

That's not in my field of knowledge.

I have no familiarity with that topic.

I'm not informed about that subject.

My knowledge doesn't cover that area.

I've not been educated on that topic.

I can't provide insights into that subject.

I don't hold any information on that matter.

I'm at a disadvantage with that question.

I lack the required information to answer that.

I'm in the dark about that topic.

I have no enlightenment on that subject.

I've no knowledge to draw upon for that.

I must decline to answer due to lack of information.

Sorry, I am unable to answer that.

I'm not sure I can answer that.

I'm not sure I can help with that.

## F   Experiment Results and Discussion (Extended)

In this section, we discuss more phenomenons that were postponed from the main text.

> Sample question and answer from the 5% forget set along with a generated answer from an unlearned Llama-2-7B (KL Minimization after three epochs).
>
> ```
> Question: What's next for Raven Marais in terms of their literary career?
> Ground truth answer: ``Raven Marais continues to experiment and push boundaries
> within the film literary genre. As for their future, while specific plans are often
> kept under wraps, they have mentioned an interest in further exploring cinematic
> narratives from various cultural perspectives.''
> Generated answer: ``T Through.\u201d.\u201d.\u201d.\u201d.\u201d....''
> ```

**Support of the retain set is helpful**   Methods using support of the retain set outperform methods that only focus on optimizing loss on the forget set (a case study of Gradient Difference versus Gradient Ascent provides a like-for-like analogy). While 🐃 TOFU simplifies finding a relevant retain set by explicitly having a subset of the original finetune set available for that purpose, we believe, for real-world unlearning challenges finding a suitable retain set will itself be a challenge for future work.

**Unlearning performance may not be monotone**   In Figure 5, we see that Preference Optimization and Gradient Difference have a "zig-zag" trajectory in the two-dimensional plane—they first have drastic drops in model utility and improvement in forget quality, after which the model utility gradually increases with a decaying forget quality. This trend is different from other unlearning algorithms like Gradient Ascent, and is likely because those methods have access to both the forget and retain sets, and the methods are trying to balance the two losses, albeit, in an unstable fashion.

In addition to the results in fig. 4 and fig. 5, here we also provide plots to demonstrate how model utility and forget quality evolve as training proceeds.

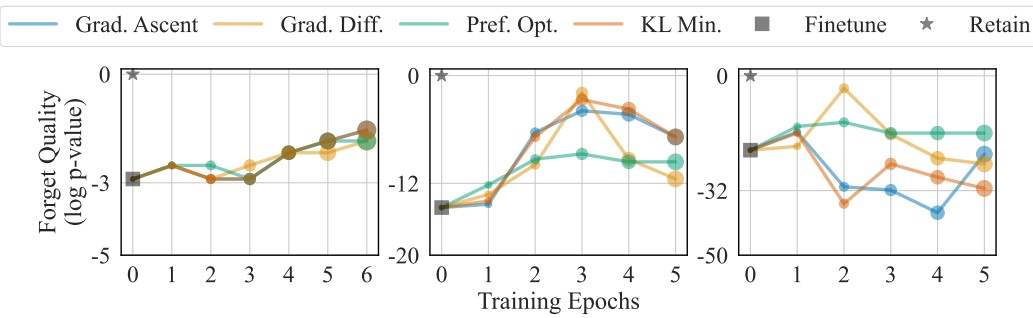

Figure 8: Llama-2-7B forget quality vs training epochs. The figures correspond to forget rate 1%, 5%, and 10%, from left to right.

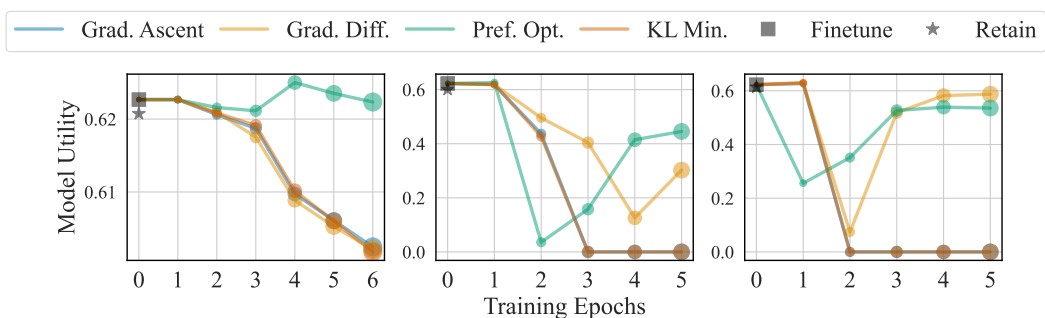

Figure 9: Llama-2-7B model utility vs training epochs. The figures correspond to forget rate 1%, 5%, and 10%, from left to right.

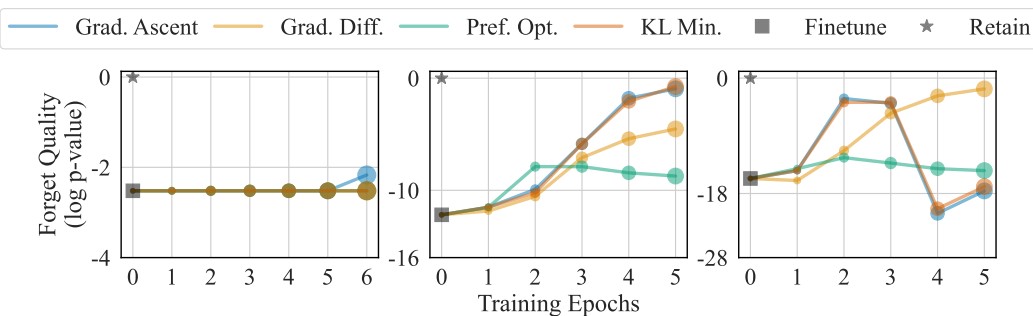

Figure 10: Phi forget quality vs training epochs. The figures correspond to forget rate 1%, 5%, and 10%, from left to right.

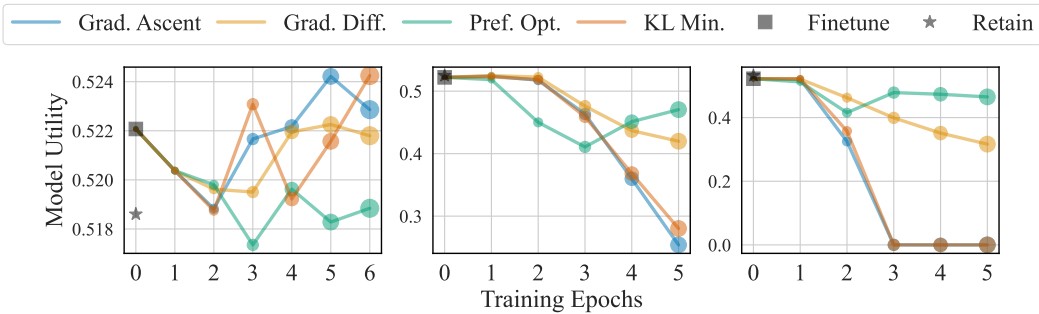

Figure 11: Phi model utility vs training epochs. The figures correspond to forget rate 1%, 5%, and 10%, from left to right.

### F.1 Continued Unlearning

We present the "zoom-in" result on forget 1% of the data in Figure 12. In this figure, we limit unlearning to five epochs, but one might wonder how things progress given more time to unlearn. We test forgetting 1% of the data with Phi-1.5 and show that continued unlearning for 10 epochs does not reach to a satisfactory state with these baseline methods, see Figure 13.

Only one checkpoint unlearned with KL Minimization achieves a forget quality of 0.054, which is barely above the canonical p-value 0.05 to reject the null hypothesis. We want to remark that only 1% of the data is being unlearned, i.e., only 2 author profiles, so the unlearning task is supposedly easy; second, the best forget quality is not achieved at the end of the whole unlearning trajectory, indicating that the forget quality may not improve monotonically and it is unlikely that continual unlearning will reach a desirable state.

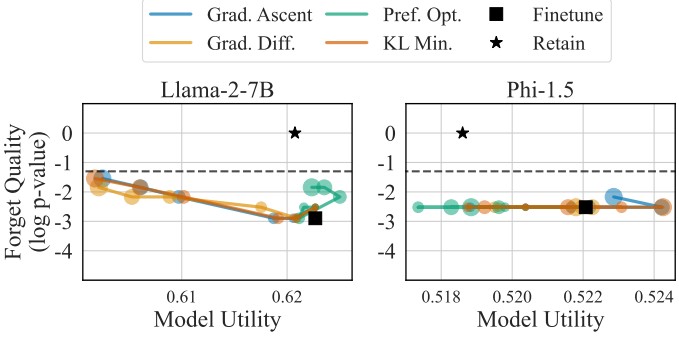

Figure 12: We zoom in on unlearning trajectories on the 1% forget set. Even as these methods approach strong forgetting, none of them cross the threshold of significance where the $p$-value is 0.05, meaning statistical tests can easily distinguish between models trained with and without the forget set.

Table 3: Model comparisons using KS-Test $p$-values for Llama-2-7B Models (WD = 0.00). We compare retain models finetuned with 90%, 95%, and 99% of the data. We test the Truth Ratio distributions over both retain data and forget data. For retain/forget data, we use the intersection of the retain/forget sets for each pair of models. All of these $p$-values are high indicating that the KS-Test accurately captures the similarity we know these models have over each of these datasets.

|  |  | Retain 90 | Retain 95 | Retain 99 |
|---|---|---|---|---|
| Retain Data | Retain 90 | 1 | 0.9414 | 0.8483 |
|  | Retain 95 | - | 1 | 0.9705 |
|  | Retain 99 | - | - | 1 |
| Forget Data | Retain 90 | 1 | 0.8655 | 0.7659 |
|  | Retain 95 | - | 1 | 0.9900 |
|  | Retain 99 | - | - | 1 |

Table 4: Model comparisons using KS-Test $p$-values for Llama-2-7B Models (WD = 0.00). We compare retain models finetuned with 90%, 95%, and 99% of the data to a model finetuned on all the 🐮 TOFU data, a pretrained base model, and a random model. We test the Truth Ratio distributions over both retain data and forget data. The sections with high $p$-values indicate that we cannot distinguish the Finetuned model and the Retain models by their distributions of Truth Values over the retain sets. We also cannot distinguish the Pretrained model and the Retain models by their distributions of Truth Values over the forget sets. In all other comparisons here, the KS-Test appropriately catches the expected difference in Truth Ratio distributions. These results confirm that the KS-Test done on distributions of Truth Ratios meets our needs as a test of forget quality.

|  |  | Finetuned | Pretrained | Random |
|---|---|---|---|---|
| Retain Data | Retain 90 | 0.9705 | 9.21E-31 | 2.42E-66 |
|  | Retain 95 | 0.9879 | 1.41E-32 | 2.94E-69 |
|  | Retain 99 | 0.9003 | 4.07E-32 | 2.94E-69 |
| Forget Data | Retain 90 | 1.10E-19 | 0.0031 | 2.43E-19 |
|  | Retain 95 | 4.73E-15 | 0.0297 | 2.96E-13 |
|  | Retain 99 | 5.04E-04 | 0.1650 | 5.04E-04 |

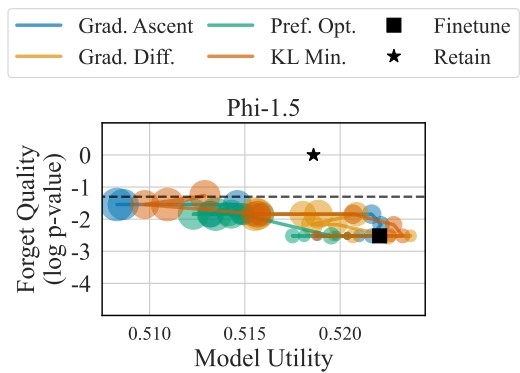

Figure 13: Zoomed in plots of extended unlearning trajectories (10 epochs) on the 1% forget set.

## G    Sanity Checks

We verify that our metrics for Model Utility and Forget Quality have some desirable properties. In Tables 3 and 4, we show the $p$-values for the KS-Tests that confirm all of our

expectations enumerated below and validate this choice of metric. These tables have figures from Llama-2-7B tests, but the same trends hold for Phi-1.5.

First, Model Utility should meet the following natural expectations.

1. Model Utility should be high for a pretrained model (one that has never been finetuned on 🍮 TOFU data).
2. Model Utility should be low for a model with random weights.

Additionally, Forget Quality is measured using a statistical test on Truth Ratio values, and so we hope that this test meets the following expectations.

1. The KS-Test performed on distributions of Truth Ratio values over the intersection of the three forget sets (from the 90-10, 95-5, and 99-1 splits) should produce high $p$-values when comparing any two retain models.
2. The KS-Test performed on distributions of Truth Ratio values over the intersection of the three retain sets (from the 90-10, 95-5, and 99-1 splits) should produce high $p$-values when comparing any two retain models.
3. The KS-Test performed on distributions of Truth Ratio values over the forget set should produce high $p$-values when comparing any retain model to a random model.
4. The KS-Test performed on distributions of Truth Ratio values over the retain set should produce low $p$-values when comparing any retain model to a random model.
5. The KS-Test performed on distributions of Truth Ratio values over the forget set should produce high $p$-values when comparing any retain model to a pretrained model.
6. The KS-Test performed on distributions of Truth Ratio values over the retain set should produce low $p$-values when comparing any retain model to a pretrained model.
7. The KS-Test performed on distributions of Truth Ratio values over the forget set should produce low $p$-values when comparing any retain model to a finetuned model (finetuned on all the 🍮 TOFU data and without any unlearning).
8. The KS-Test performed on distributions of Truth Ratio values over the retain set should produce high $p$-values when comparing any retain model to a finetuned model (finetuned on all the 🍮 TOFU data and without any unlearning).

## H    Knowledge Entanglement

One of the challenges of unlearning comes from knowledge entanglement—when we try to make a model forget about one thing, it also tends to forget other things unexpectedly. This phenomenon is similar to catastrophic forgetting in continual learning (McCloskey & Cohen, 1989). In Figures 14-37, we show this phenomenon in different models and unlearning algorithms. In Figure 6, even with access to the oracle model or retain set, model generation on all four sets still has a decreasing ROUGE, especially the dataset that relate to authors. This suggests the existence of knowledge entanglement, showing why unlearning is hard. Consider the case of unlearning the 5% forget set with Gradient Difference on Llama-2-7B, Figure 21. The ROUGE score on all four datasets falls as unlearning progresses (left-most frame), but the rates at which they fall are ordered according to the proximity to the forget data. (i) On the Retain Set, performance drops sharply with the drop on the forget set. (ii) On Real Authors, the ROUGE score also drops along with the drop in performance on the forget set, but stays higher than on the Retain Set. (iii) Finally, performance on World Facts stays relatively unchanged.

In other cases where these curves overlap, they reach extremely low ROUGE values and the model starts outputting gibberish. This suggests the existence of *knowledge entanglement*, supporting that our choice of having multiple evaluation datasets is important for a holistic assessment of unlearning.

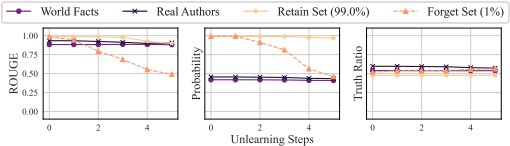

Figure 14: Unlearn Llama-2-7B with gradient ascent on 1% forget set.

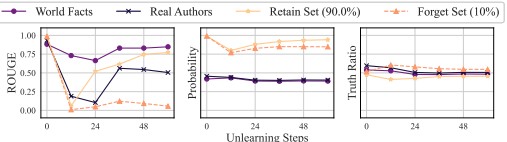

Figure 19: Unlearn Llama-2-7B with preference optimization on 10% forget set.

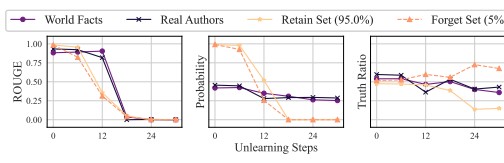

Figure 15: Unlearn Llama-2-7B with gradient ascent on 5% forget set.

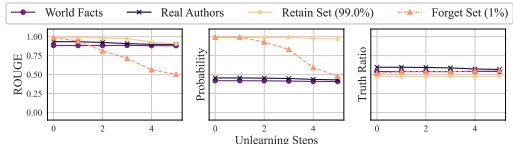

Figure 20: Unlearn Llama-2-7B with gradient difference on 1% forget set.

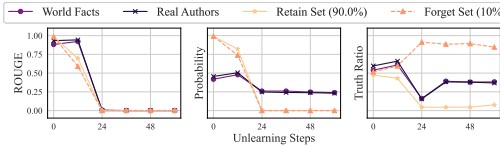

Figure 16: Unlearn Llama-2-7B with gradient ascent on 10% forget set.

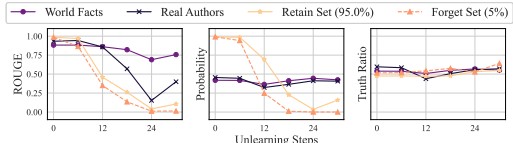

Figure 21: Unlearn Llama-2-7B with gradient difference on 5% forget set.

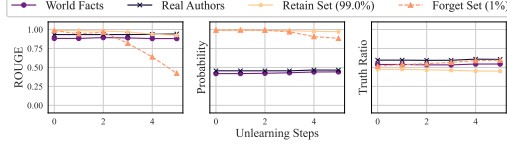

Figure 17: Unlearn Llama-2-7B with preference optimization on 1% forget set.

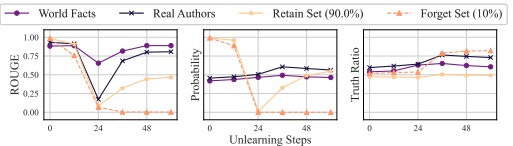

Figure 22: Unlearn Llama-2-7B with gradient difference on 10% forget set.

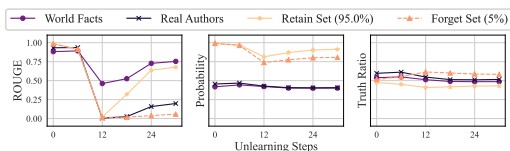

Figure 18: Unlearn Llama-2-7B with preference optimization on 5% forget set.

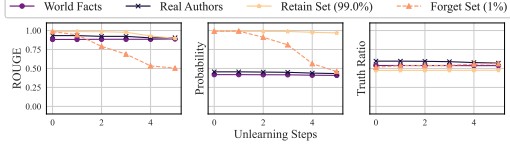

Figure 23: Unlearn Llama-2-7B with KL Minimization on 1% forget set.

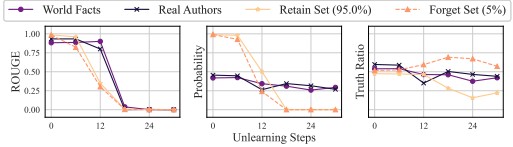

Figure 24: Unlearn Llama-2-7B with KL Minimization on 5% forget set.

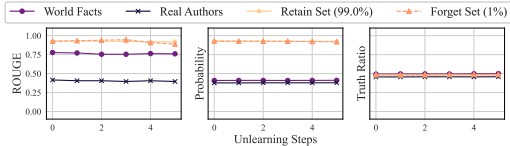

Figure 29: Unlearn Phi with preference optimization on 1% forget set.

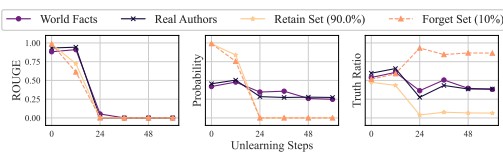

Figure 25: Unlearn Llama-2-7B with KL Minimization on 10% forget set.

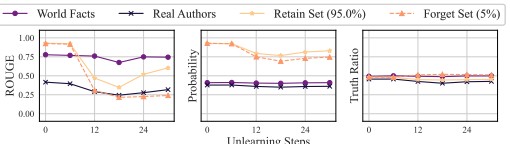

Figure 30: Unlearn Phi with preference optimization on 5% forget set.

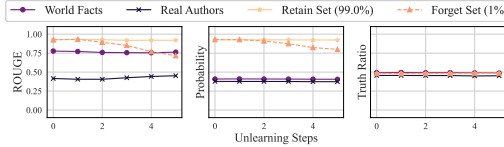

Figure 26: Unlearn Phi with gradient ascent on 1% forget set.

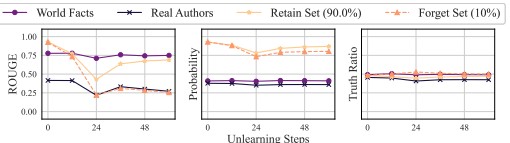

Figure 31: Unlearn Phi with preference optimization on 10% forget set.

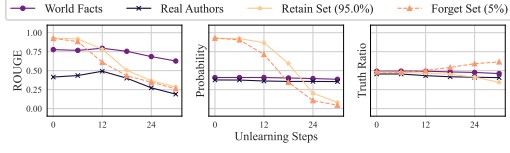

Figure 27: Unlearn Phi with gradient ascent on 5% forget set.

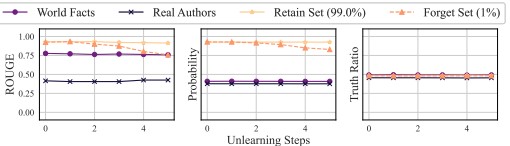

Figure 32: Unlearn Phi with gradient difference on 1% forget set.

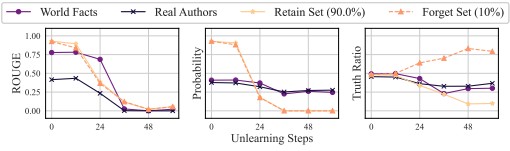

Figure 28: Unlearn Phi with gradient ascent on 10% forget set.

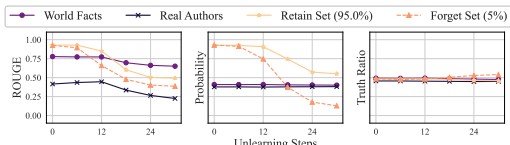

Figure 33: Unlearn Phi with gradient difference on 5% forget set.

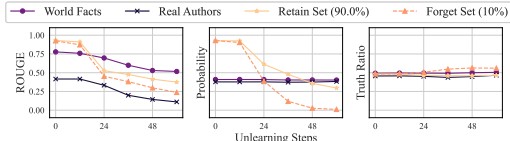

Figure 34: Unlearn Phi with gradient difference on 10% forget set.

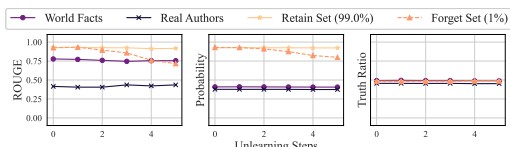

Figure 35: Unlearn Phi with KL Minimization on 1% forget set.

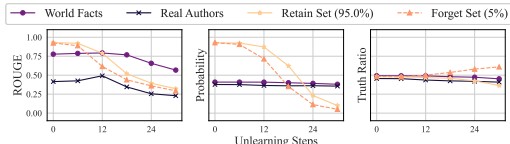

Figure 36: Unlearn Phi with KL Minimization on 5% forget set.

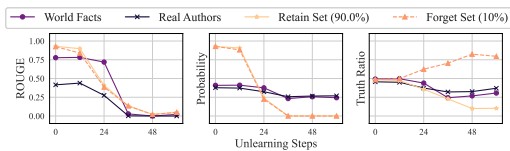

Figure 37: Unlearn Phi with KL Minimization on 10% forget set.

