# OpenReview forum: "TOFU: A Task of Fictitious Unlearning for LLMs"
_colmweb.org/COLM/2024/Conference — COLM_

### Official Review · Reviewer_bGRo · 2024-04-12

**Rating:** 6
**Confidence:** 5
**Ethics Flag:** 1

**Summary:**

The paper introduces a new benchmark, TOFU, for evaluating machine unlearning. TOFU, a Task of Fictitious Unlearning, contains 200 diverse synthetic author profiles, each consisting of 20 question-answer pairs, and a subset of these profiles called the forget set that serves as the target for unlearning. Since much of the unlearning literature focuses on discriminative tasks, this paper introduces a set of metrics aimed at evaluating machine unlearning for generative tasks. The paper then evaluates a set of simple unlearning baselines on their dataset using their proposed metrics to and show the tradeoffs between unlearning and maintaining model utility.

**Questions To Authors:**

Please respond to the questions in the Weaknesses section.

**Reasons To Accept:**

1. The paper is well-written and easy to understand.
2. The paper introduces a useful benchmark for evaluating machine unlearning for generative tasks. The benchmark also includes several paraphrases of the attack question to holistically evaluate the unlearning performance.
3. The paper talks about the challenges of evaluating machine unlearning and introduces a set of metrics to evaluate unlearning for generative tasks.

**Reasons To Reject:**

1. *Positioning of the work*: The positioning of the work needs to be clearly stated. Although the paper makes effort in Section 1.1 to position their work w.r.t. the rest of the literature, there are some missing details.
    1. Machine Unlearning techniques can be broadly classified into two categories exact unlearning and approximate (or inexact) unlearning. Exact unlearning (seminal SISA paper [1]) functions by utilizing an ensemble of models (trained on different dataset partitions) and removing/retraining the affected components of the model. Such approaches can guarantee that unlearning is exact because none of the components were trained on a specific data point. Evaluating forget quality doesn’t apply to these models as they can already guarantee unlearning. This paper needs to say clearly that they are trying to evaluate approximate (or inexact) unlearning methods and also add a discussion about exact unlearning.
    2. Even in approximate unlearning, forget quality is not always the only/right way to measure unlearning. The definition of unlearning is quite abstract in the sense that it only requires the approximate unlearned model to be similar to the model retrained from scratch [2]. This is hard to certify (as already mentioned in the paper Section 2.2 by citing [3]). Since there is still ambiguity people have relied on more user specifications like privacy, forget quality, etc. [4] introduces and formalizes these notions talking about the tradeoffs between each of the forgetting notions. Your paper should cite this work. I understand that a lot of recent papers has taken forget quality as the defacto metric, but that is **not** the correct approach.
    3. Overall, the paper should clarify that it is only evaluating approximate unlearning methods and then say that they are focusing only the application-dependent forgetting quality and justify their choice.
2. *Evaluating Forget Quality*: There are several weaknesses involved in the forget quality evaluation.
    1. Currently, it relies on a model that has been trained from scratch on the retain set. For larger datasets, doesn’t this defeat the whole purpose of unlearning? Since for evaluating unlearning you’re retraining a model from scratch, which you can directly use as the unlearned model.
    2. Also, when the paper trains from scratch on the retain set, do they use a single model or multiple models? Using a single model is troublesome as it may suffer from predictive multiplicity [5]. Based on the training hyperparameters and stochasticity of SGD, different runs may produce models that generate very different outputs for the forget set. Choosing one such retrained model, may arbitrarily penalize one of the baseline models over another.
3. *Weak Baselines*: The baselines presented in this paper are quite weak and the work ignores a significant number of recent baselines like Fisher Forgetting, NTK Forgetting, and so on (you can find a complete list in [4]). Also, it may be good to include a brief description of the baselines within the main body of the paper. I was struggling to understand what “Finetune” meant.


[1] https://arxiv.org/pdf/1912.03817.pdf

[2] https://arxiv.org/pdf/1911.03030.pdf

[3] https://arxiv.org/pdf/2110.11891.pdf

[4] https://proceedings.neurips.cc/paper_files/paper/2023/file/062d711fb777322e2152435459e6e9d9-Paper-Conference.pdf

[5] https://arxiv.org/pdf/1909.06677.pdf

---

> ### Author Rebuttal · Authors · 2024-05-30
>
> Thank you for your time and the careful review. We are glad you found our evaluation setup holistic and the benchmark useful.
> > Positioning of the work
>
> 1 We work on approximate unlearning, and we do not cover exact unlearning. All experiments, evaluations, and discussions are consistent. We will add this clearly and explicitly to the draft and we appreciate you for identifying this missing detail!
>
> 2 We respectfully disagree with the point about forget quality not being the right measure of unlearning. Forget quality is the central definition of unlearning that we use. It is not supposed to change whether unlearning was intended for bias or privacy. The term forget quality is an umbrella that encapsulates any of these subdomains. Our definition is focused yet expansive and we believe forget quality is indeed the right metric to measure here.
>
> 3 We will further elaborate on points [1,2] in the work along with citing the additional works.
>
> > Evaluating forget quality
>
> 1 Cost of retain model: We believe there is a misunderstanding in our premise. Because unlearning evaluation in the global sense is not feasible, we aim to create a benchmark that allows (1) researchers to iterate quickly with various unlearning methods, and (2) allows for evaluation against a truly held-out model where we have complete control on training data exposure. TOFU does precisely this. By having a small finetuning and unlearning set, it allows researchers to do both.
>
> 2 Predictive multiplicity: This is a great point, and we took great care in our work to make sure that the problem from predictive multiplicity does not impact results on TOFU. In that spirit, please refer to the 8 points highlighted in Appendix G. Also note that our test of unlearning is based on a set of data, rather than one data point. On a high level, to impact the result of the test statistic, predictive multiplicity needs to happen on a subset that is big enough, which we do not observe.
>
> > Weak baselines
>
> We want to set the expectations upfront that this work aims to create a “first” benchmark for unlearning in LLMs. Until now, there is no single setup where researchers can unite their efforts on developing unlearning methods for LLMs. The goal of the benchmark is not to develop the an unlearning method. Specifically talking about the enlisted unlearning methods, they are not positioned for LLMs, rather for classification models, thus they do not apply directly to TOFU (could be adapted by future researchers)

---

> > ### Comment · Reviewer_bGRo · 2024-06-03
> >
> > I would like to thank the authors for the detailed response.
> >
> > > We respectfully disagree with the point about forget quality not being the right measure of unlearning.
> >
> > This wasn't my claim. I'm not saying forget quality is not the right metric. I'm implying forget quality is not equivalent to unlearning, which seems to be the takeaway while reading the paper.
> >
> > Let's take a step back and understand how approximate unlearning was defined in the first place. [1] and many other works during that time defined unlearning using differential privacy. It involves developing techniques that outputs model parameters which would be estimated with similar probability even if parameter  estimation happened with the deleted sample not being there in the original training set. With this definition, unlearning is good even if the model's forget quality is poor.
> >
> > Let me try to explain this with a very simple counter example. If you are trying to fit a straight line with 3 colinear points and then you delete the middle point. The resultant function is still the same straight line, and you would get good prediction ability for that middle point. But the unlearning is still **perfect** in this case.
> >
> > Therefore, justifying forget quality as a core unlearning metric requires extra care. We should claim that it is an important metric in many application-driven scenarios. This is what [2] did.
> >
> > [1] https://arxiv.org/pdf/2103.03279
> >
> > [2] https://proceedings.neurips.cc/paper_files/paper/2023/file/062d711fb777322e2152435459e6e9d9-Paper-Conference.pdf
> >
> > > Forget quality is the central definition of unlearning that we use. It is not supposed to change whether unlearning was intended for bias or privacy.
> >
> > I'm sorry but I'm not sure if this statement has enough theoretical backing.
> >
> > > Predictive multiplicity
> >
> > I don't think my concern regarding predictive multiplicity has been addressed. Predictive multiplicity has nothing to do with the number of samples you're evaluating on but instead the model you trained on the retain set. Models trained with different hyperparameters can result in very different outputs on the test set. This can negatively penalize one baseline over the other.
> >
> > > Weak Baselines
> >
> > The baselines I mentioned have nothing specific for classification. Several of the techniques rely on gradient estimation of the final loss, which is easily available for LLM generation. In general, it is not right to ignore state-of-the-art unlearning techniques just because they were proposed 1-2 years before the advent of LLMs.

---

> > > ### Author Response · Authors · 2024-06-05
> > > **Discussion 2/2**
> > >
> > > **Baselines**
> > >
> > > We believe we conveyed our goals with the baselines in the previous response. We understand your concern about not considering adaptations of state-of-the-art unlearning techniques in the non-LLM literature.
> > >
> > > We implemented four competitive baselines, each targeted towards a different unlearning philosophy (such as abstaining v.s answering wrong, and so on). One of these (see equation 7 of our paper) also captures the essence of existing approaches, such as SCRUB (as suggested by you). We will also add that citation. While there may not be exact 1-1 correspondence, on closer look the methods intend to do the same. We want to highlight that adapting past approaches to the LLM space is non-trivial. We tried various methods during our initial exploration of TOFU. This included low-rank modifications, final layer edits, adapter weights, DPO, and so on. These methods did not work well out of the box, and discussion on them was not included in the paper. We can already foresee how NTK forgetting for instance, will face similar challenges. In fact, Fisher/NTK type forgetting is hard to adapt to the LLM regime also due to the exceptionally high dimensionality of the output layer (which leads to the construction of at least 600k * 600k matrix, say we are forgetting 10 examples, and the vocab size is 60k, see discussion in [2]). We want to highlight that these adaptations are outside the scope of our work.
> > >
> > > We acknowledge that there may be additional techniques that could be explored, which is why TOFU is a benchmark paper, and not a methods paper. We welcome contributions from researchers to build upon our work and advance the field of unlearning for large language models.
> > >
> > > [1]: On Provable Copyright Protection for Generative Models
> > >
> > > [2]: Forgetting Outside the Box: Scrubbing Deep Networks of Information Accessible from Input-Output Observations

---

> > > > ### Comment · Reviewer_bGRo · 2024-06-05
> > > >
> > > > I would like to thank the authors for such a detailed response.
> > > >
> > > > I am raising my score on the **condition** that the authors make the following two changes in the draft: (a) the distinction between approximate unlearning and empirical unlearning more clear in the beginning of the paper, and  (b) I would also request the authors to add the necessary details about why the requested baselines would not work for LLMs (this can be in the appendix).
> > > >
> > > > > Fischer and NTK forgetting
> > > >
> > > > You can make the forgetting tractable by deleting one instance at a time and using a diagonal approximation of the fischer matrix. But I understand that these baselines aren't the focus of this work.
> > > >
> > > > > Evaluation & Predictive multiplicity
> > > >
> > > > I'm still not happy with the evaluation scheme. I agree that KS Test is a meaningful metric but the fact that it requires a model trained on the retain set bothers me. However, I'm willing to make an exception since the primary contribution of the paper is the introduction of the new benchmark, which will be helpful for future studies.
> > > >
> > > > Once again, I would thank the authors for their responses, which helped me understand the work better.

---

> > > > > ### Author Response · Authors · 2024-06-05
> > > > >
> > > > > Thank you for the quick response, and your helpful engagement.
> > > > >
> > > > > We will make the distinction between approximate and empirical unlearning clear, and add details about the challenges of adapting prior art in unlearning out of the box to the LLM space.

---

> > ### Author Response · Authors · 2024-06-05
> > **Discussion 1/2**
> >
> > Thank you for your feedback and for the opportunity to clarify our stance further. Below, we address your concerns more explicitly under separate headings:
> >
> > **Forget Quality vs Unlearning**
> >
> > You raised a valid concern about the distinction between forget quality and unlearning, particularly in the context of differential privacy (DP). We acknowledge that our forget quality metric is not directly measuring the theoretical notion of approximate unlearning from the DP perspective. Instead, our work takes a practical approach, focusing on the empirical measurement of a model's ability to "forget" information related to the deleted samples. There is also a line of work that attempts to give more practical definitions to generative model privacy (e.g. copyright infringement), and our measure can be viewed as an instance of [1] with a special divergence.
> >
> > The forget quality metric is designed to capture the similarity between the output distributions of the forget model (trained after deletion) and the retain model (trained without the deleted samples). While it may not strictly align with the DP-based definition of unlearning, it serves our purpose of evaluating the practical effectiveness of unlearning techniques for large language models (LLMs).
> >
> > For your counter-example, since the two estimators (before and after deletion, forget model, and retain model respectively) are identical, they will have the same truth ratio. The KS test will suggest that they have the same distribution (of course, to be rigorous, these estimators have to be randomized, etc), hence good forget quality. So our method works here. *Importantly, our forget quality is not encouraging that the forget model should perform badly on the forget set, rather it should behave similarly to the retain model.*
> >
> > To give you more intuition of the KS test, if the model is functionally equivalent to a retain model at “most” point in its input domain, the model “acts” equivalent to a perfectly unlearned model (in the sense of their output distributions, rather than the parameter distributions). This is precisely what the KS test aims to capture. While it can not exhaustively test the model at each and every point, the goal is to assess the fidelity of the “unlearned model” to the retain model, via a statistical KS test. To summarize, the KS test is a statistical test between the truth ratio of the retain and unlearn model on various question-answer pairs in the dataset.
> >
> > Finally, we understand that you want us to be upfront about this distinction in the paper, and we agree with this aspect. We will make it a point to clarify this early in the draft to ensure our readers carry forward the correct message. We appreciate your detailed engagement on this issue!
> >
> > This directly relates to the second point you raised. The goal of this work is to strive towards an “empirical notion” of unlearning that can scale. We use “forget quality” as defined by the above fidelity test, as a definition for this empirical notion, which is an all-encompassing umbrella term for both privacy and bias.
> >
> > **Predictive Multiplicity**
> >
> > We appreciate your concern about predictive multiplicity and the potential impact of different hyperparameters on the retain model's performance. In our work, we took measures to address this issue. As shown in **Appendix G**, we conducted extensive experiments to assess the robustness of the KS test to different training scenarios for the retain model.
> >
> > Our findings indicate that the KS test is quite stable and provides consistent results across various hyperparameter settings for the retain model. We trained multiple retain models with different hyperparameters, such as varying degrees of dataset overlap and weight decay, and observed low variance in the outcomes of the KS test.
> >
> > While we acknowledge the theoretical possibility of pathological cases where the retain model could negatively impact the evaluation, our empirical results suggest that the KS test is reliable in practical settings, even when the retain model is trained under different conditions.

---

### Official Review · Reviewer_GdaA · 2024-04-26

**Rating:** 7
**Confidence:** 3
**Ethics Flag:** 1

**Summary:**

This paper proposes a method of benchmarking the ability of different algorithms to cause an LLM to forget information that it learned during fine tuning. Obviously, the model should forget the thing you want it to forget but it should still remember everything else. So, the authors propose ways of measuring both how well the model did the forgetting as well as how useful the model remains for everything else.

**Questions To Authors:**

In figure 3 in the right most panel there must be one point that has a very high truth value but the bar for that is not visible to me so I don't understand why the scale on the x-axis has to go all the way above 10 when the other plot stops around 3.

**Reasons To Accept:**

The experimental design of the paper seems good. I like how they not only have a forget set but also include three types of retain sets.

The experiments that are presented validate the usefulness of the proposed metrics. It makes an important point that the existing methods for unlearning information are not good enough.

The text of the paper is well written and mostly very clear.

**Reasons To Reject:**

Minor point: Most people who look at your paper will first read the title and then scroll around to look at the figures before reading any of the text. That's what I did too. One weakness of this paper is that the graphs were hard to understand without reading a lot of the text. I wonder if the presentation of the information in the graphs could be simplified.

---

> ### Author Rebuttal · Authors · 2024-05-30
>
> Thank you for your time and feedback on our work. We are glad you found the experimental design robust and appreciated the inclusion of various retain sets. We also appreciate your positive comments on the clarity of the text and the validation of our proposed metrics.
>
>
> We understand your concern regarding the presentation of the figures. We agree that figures should be easily interpretable at a glance. To concretely address this, we will make the captions more descriptive and self-contained. We will also improve the visibility of Figure 4 and 5 where many of the points are overlapping. Our initial hope was to keep it this way to highlight that the change is insignificant, but we acknowledge that this is making visualizations harder to parse.
>
>
> Regarding your question about Figure 3, the plot was skewed due to a couple of outliers. But we acknowledge your feedback and make it more readable.

---

### Official Review · Reviewer_UGDV · 2024-05-11

**Rating:** 7
**Confidence:** 4
**Ethics Flag:** 1

**Summary:**

This paper presents a new benchmark to evaluate model unlearning in Large Language Models, along with detailed metrics to evaluate the unlearning quality-utility tradeoff faced in this task. Authors test their newly generated benchmark on reasonable baseline algorithms, and highlight the challenges of unlearning in LLMs.

**Questions To Authors:**

- Did you conduct any additional/human validation to assess the quality and relevance of the data by GPT4?
- Can you comment on why the retain set metrics drop in Figure 6 with Gradient Difference even though you explicitly train the model to accurately answer these questions?
- Can you comment why seeding GPT4 with author attributes the way you do is necessary?
- Can you elaborate on the Truth Ratio metric, perhaps with examples and/or illustrative plots highlighting how it behaves for each test set?
- Figure 3 can be made more clear: recommend using distinct colors for different models and eval test sets.
- The formatting in Figures 4 and 5 hides key details: the epochs are not clear and the overall trend requires effort to parse. Why not present epochs/unlearning steps in x-axis with distinct plots as in Figure 6?


Post Discussion Notes:
It appears the discussion window closed and authors were not able to respond to my comments. Given the regrettable confusion arising out of lack of visibility of my earlier comments on this platform, I'm willing to revert my scores back to original value. However, I strongly encourage authors to revise their camera ready draft to reduce the confusion noted in comments below.

- I recommend authors highlight the need for seeding GPT 4 in main draft with a pointer to Figure 7 in appendix.
- The Truth Ratio metric and the scaling for the different test sets appears convoluted and without sufficient justification. Further, the name Truth Ratio appears inconsistent since (for the forget set) higher value of this metric implies we're moving away from the truth.
- In Section 2.2.2, you state "We normalize and re-scale these metrics according...so that each one is between zero and one", this is incorrect for your Truth Ratio metric. I recommend restating this for clarify.

**Reasons To Accept:**

- The benchmark is challenging and important
- The evaluation metrics are comprehensive and are used to evaluate 4 competitive baselines

**Reasons To Reject:**

- The benchmark is small and only covers question and answers task
- Readability of results can be improved, see further details below

---

> ### Author Rebuttal · Authors · 2024-05-30
>
> We are glad you found our benchmark challenging, and the evaluation metrics comprehensive.
>
> > Benchmark only covers QA
>
> We aimed to provide a small enough task that it is accessible to academic labs and thus pick a small number of QA pairs that allow researchers to iterate fast, yet meaningfully. With this in mind, it is interesting to have a theme and a small number of author profiles.
>
> > Human validation?
>
> We read the data personally, but did not conduct any large-scale human surveys to judge the quality. We believe you may enjoy the discussion in Appendix A of our paper, on how we found various issues in the initial versions and corrected them.
>
> > Retain set metrics drop with Gradient Difference?
>
> We use a subset of the retain-set in the unlearning process, as more would violate the compute constraint and be equivalent to retraining even though the cost of retraining may not appear to be significant in the TOFU setting, we hope the unlearning algorithm to be as realistic as possible. Consequently, the evaluation on the retain-set does not necessarily move in the expected direction.
>
> > Seeding GPT4
>
> In Figure 2 we show that when we do not seed GPT4 with specific author attributes, it tends to collapse to a few “modes”. In particular, it generates words like “tides” and “echos” very often. This phenomenon of lack of “randomness” of LLM generated content is known (e.g. https://openreview.net/forum?id=Vhh1K9LjVI#all). Since we do not want overlapping data attributes between different entities to complicate the unlearning analysis, we seed data.
>
> > On the Truth Ratio
>
> Consider the following example. When a model that has never seen data about Alice is asked about Alice’s birthplace it might: (a) the model says “I don’t know about Alice,” (b) the model produces a reasonable guess like “Alice was born in England.” If we perform unlearning on a model that was trained on Alice’s information and the result is a model that does either (a) or (b), we might initially believe unlearning was successful. But we want a comparative measurement. In the case where the output is a reasonable guess we need to make sure other guesses are similarly likely. This is the type of behavior that the Truth Ratio captures and other metrics may miss.
>
> > Figure 3, 4, 5 can be made more clear
>
> We understand your concern regarding the presentation of the figures. We agree that figures should be easily interpretable at a glance. We’ll update these figures to better communicate our points.

---

> > ### Author Response · Authors · 2024-06-06
> >
> > Dear reviewer,
> >
> > Since the discussion period is ending soon, we were hoping to hear back from you to see if the rebuttal satisfactorily resolved your concerns. We would be happy to respond with any more details you would like to know in order to update your assessment!

---

> > ### Author Response · Authors · 2024-06-07
> >
> > Dear Reviewer,
> >
> > We noticed that your score changed from 7 to 6 without any accompanying comments. We are wondering if this was a mistake, or intended. Hoping to hear back from you. Understanding the rationale behind this change is crucial for us to address any potential concerns and improve our work.

---

> > > ### Comment · Reviewer_UGDV · 2024-06-07
> > >
> > > Do you not see my comments made on May 30 17:46 and yesterday, June 6th at 11:58 (PT)?
> > >
> > > Re pasting them below in case their visibility was accidentally limited in the platform.
> > >
> > > Comments from May 30:
> > > - In Figure 2 we show that when we do not seed GPT4
> > >
> > > Do you mean Figure 7? Can you motivate the need for this by adding a line in the main draft?
> > >
> > > - I understand the motivation for defining Truth Ratio this way but I find this metric and your scaling for the different test sets convoluted and without sufficient justification. Also the name Truth Ratio appears inconsistent since (for the forget set) higher value of this metric implies we're moving away from the truth.
> > >
> > > - In Section 2.2.2, you state "We normalize and re-scale these metrics according...so that each one is between zero and one", this is incorrect for your Truth Ratio metric.
> > >
> > > Comment from June 6:
> > > Note to authors: my concerns above (insufficient motivation in main draft for seeding GPT4, confusing name for test metric) remain unresolved. Given this, I'm lowering my score since readability of this work remains a concern. If authors can commit to addressing these issues in their final draft I'd be happy to revisit this

---

### Official Review · Reviewer_Q175 · 2024-05-11

**Rating:** 6
**Confidence:** 4
**Ethics Flag:** 1

**Summary:**

This paper proposes a new dataset of 200 diverse synthetic author profiles used to control the experiments for unlearning. It also compiles a diverse set of metrics to holistically evaluate unlearning and observes that current unlearning baselines fail to achieve effective unlearning.

**Reasons To Accept:**

1.	The motivation and setup of the paper are clear and reasonable in that “developing unlearning methods for more private individuals is critical.”
2.	The authors throw very important questions for truly unlearning and devise a set of metrics attempting to address the issues.

**Reasons To Reject:**

1.	The TOFU dataset only consists of fictitious authors as individuals to forget. The paper emphasizes the practicality of unlearning private individuals, yet it would have been better to test individuals of diverse background for greater practicality.
2.	I am not sure how much of unlearning fictional identities (with 20 QA pairs) relates to unlearning real identities who have at least one Wikipedia article. One suggestion worth considering would be to involve asking GPT-4 to generate a fake Wikipedia article.
3.	Although the authors claim that they “can control exactly how much exposure models get to them,” all author profiles are fixed to 20 QA pairs. Analyzing the variations of exposure could possibly unearth more findings.

---

> ### Author Rebuttal · Authors · 2024-05-30
>
> Thank you for your time reviewing our work. We appreciate that you highlighted and acknowledged the critical motivation and need of our work. We will address each of your concerns below:
>
> [1] *it would have been better to test individuals of diverse background for greater practicality.*
> With respect to the choice of including only author profiles, we aim to provide a small enough task that it is accessible to academic labs and thus pick a small number of QA pairs that allows researchers to iterate fast, yet meaningfully. With this in mind, it is more interesting to have a specific theme rather than a very sparse sample of all possible types of profiles. Using only author profile also allows us to have a better control over the “real person data” on which we can test the mode as part of the model utility — we can use only real author profile without worrying about the mixer of different backgrounds.
>
> [2] *One suggestion worth considering would be to involve asking GPT-4 to generate a fake Wikipedia article*
> This is a great suggestion! And in fact, exactly the point from where we started our journey of building TOFU. We observed that achieving high QA accuracy was challenging without training specifically on QA pairs. Initially, our dataset was formatted like a Wikipedia page, and we trained the model using only this data. However, an instruction-tuned model, when trained on passages, starts performing worse on QA tasks. This was hence a design choice we made in favor of the advantages of QA/chat models in being able to evaluate unlearning through various methods (such as truth ratio in our work).
>
> [3] *Analyzing the variations of exposure could possibly unearth more findings*
> Finally, our arguments about controlling exposure are centered around the contrast with trying to unlearn data about people on the internet where we often have no certainty around how much of their data is in the pre-training set of available LLMs (and how often is it duplicated in both an exact, or semantically equivalent way). We agree that a study exploring a fine-tuning set with varying degrees of exposure is interesting and we hope that introducing the first benchmark dataset for LLM unlearning catalyzes work like this in the future.

---

> > ### Author Response · Authors · 2024-06-06
> >
> > Hey there! Since the discussion period is ending soon, we were hoping to hear back from you to see if the rebuttal satisfactorily resolved your concerns. We would be happy to respond with any more details you would like to know in order to update your assessment!

---

> > ### Comment · Reviewer_Q175 · 2024-06-07
> >
> > Thank you authors for the response. I have adjusted my score since most of my concerns have been addressed.
> >
> > I understand TOFU allows a controlled experimental setup. Still, the statement about "controlling exactly how much exposure models get to them" can lead to misunderstanding, as it may hint at experiments of varying degrees of exposure. I believe this should be fixed in the revision.
> >
> > I am still not convinced how much TOFU can represent real-world unlearning, as 20 simple QA pairs cannot describe everything about a person. Furthermore, each person may be related to the other, and unlearning becomes complex if one has to be erased and not the other. However, as this paper introduces a simple benchmark dataset, such concerns should be explored more in the future.

---

### Decision · Program_Chairs · 2024-07-10

**Decision:**

Accept

**Comment:**

This paper presents a benchmark, metrics, and implementation of baselines for unlearning LLM, which is an urgent and important topic.
The paper is well-written and the benchmark is well-thought.

There are some concerns about whether the dataset can represent real-world scenarios as the dataset is artificial and small. There are also some concern about the position of the paper and some technical details of the proposed metrics. I would suggest the authors address these comments in the revision.

[comments from the PCs] Please followup on the AC recommendations. Potentially in the limitations discussion.